# Both Motor and Non-Motor Fluctuations Matter in the Clinical Management of Patients with Parkinson’s Disease: An Exploratory Study

**DOI:** 10.3390/jpm13020242

**Published:** 2023-01-29

**Authors:** Matteo Carpi, Mariangela Pierantozzi, Stefano Cofano, Mariana Fernandes, Rocco Cerroni, Francesca De Cillis, Nicola Biagio Mercuri, Alessandro Stefani, Claudio Liguori

**Affiliations:** 1Department of Systems Medicine, University of Rome “Tor Vergata”, 00133 Rome, Italy; 2Department of Psychology, “Sapienza” University of Rome, 00185 Rome, Italy; 3Parkinson’s Disease Unit, University Hospital of Rome “Tor Vergata”, 00133 Rome, Italy; 4Department of Clinical Science and Translational Medicine, University of Rome “Tor Vergata”, 00133 Rome, Italy; 5Neurology Unit, University Hospital of Rome “Tor Vergata”, 00133 Rome, Italy

**Keywords:** Parkinson’s disease, non-motor fluctuations, non-motor symptoms, motor impairment

## Abstract

Non-motor symptoms (NMS) characterize the Parkinson’s disease (PD) clinical picture, and as well as motor fluctuations, PD patients can also experience NMS fluctuations (NMF). The aim of this observational study was to investigate the presence of NMS and NMF in patients with PD using the recently validated Non-Motor Fluctuation Assessment questionnaire (NoMoFa) and to evaluate their associations with disease characteristics and motor impairment. Patients with PD were consecutively recruited, and NMS, NMF, motor impairment, motor fluctuations, levodopa-equivalent daily dose, and motor performance were evaluated. One-third of the 25 patients included in the study (10 females, 15 males, mean age: 69.9 ± 10.3) showed NMF, and patients with NMF presented a higher number of NMS (*p* < 0.01). Static NMS and NoMoFa total score were positively associated with motor performance assessed with the Global Mobility Task (*p* < 0.01 and *p* < 0.001), and the latter was also correlated with motor impairment (*p* < 0.05) but not with motor fluctuations. Overall, this study shows evidence that NMF are frequently reported by mild-to-moderate PD patients and associated with an increased number of NMS. The relationship between NoMoFa total score and motor functioning highlights the importance of understanding the clinical role of NMS and NMF in the management of PD patients.

## 1. Introduction

The clinical importance of non-motor symptoms (NMS) in Parkinson’s disease (PD) has been extensively documented in the recent past, with many studies highlighting their high frequency and their implication for the disease’s treatment and management [1,2,3,4]. The wide range of NMS [3,4] may be present from the PD premotor stage, also before the onset of the characteristic motor symptoms of PD [1,5,6], and naturally become more prevalent and disabling over the progression of the disease [3]. NMS burden can correlate with the severity of motor impairment, affecting patient’s well-being and quality of life [7,8,9]. Moreover, the prevalence of NMS is very high in motor-fluctuating PD patients independently of the global motor impairment [10]. Consistently, fluctuations in PD motor symptoms have been frequently observed in association with chronic levodopa treatment [11,12,13]. Similarly, NMS may also present fluctuations, and indeed, non-motor fluctuations (NMF) are highly prevalent in PD patients and apparently yield a major impact on disease-related disability [14,15,16]. However, although some patients can perceive their occurrence as more disturbing than motor fluctuations [15,17], NMF are frequently underestimated in clinical settings [14].

Previous research classified NMF in three broad groups, namely cognitive/psychiatric (depression, anxiety, psychotic symptoms, mania, apathy, fatigue, and cognitive dysfunction); autonomic (including sweating, facial flushing, dysphagia, constipation, urinary frequency, blood pressure changes, and dyspnea); and sensory (such as pain, akathisia, and numbness) [18], with cognitive and psychiatric symptoms being the most frequent and disabling and cognitive dysfunction being distinct from age-related cognitive decline and mainly involving executive functions [19]. Typically, fluctuating NMS mostly occur in OFF motor states [14] although they might also be present in patients without motor fluctuations [20,21].

Considering the challenges in identifying NMF, the recently validated Non-Motor Fluctuation Assessment questionnaire (NoMoFa) [22] is the only patient-rated instrument specifically developed to detect the presence and the intensity of NMF. Using NoMoFa to identify and quantify NMF in patients with PD, this study aimed at (1) evaluating the presence and the frequency of fluctuating and non-fluctuating non-motor symptoms in a sample of PD patients and (2) exploring the relationships between NMF and sex, disease severity, and motor impairment assessed with a standardized task.

## 2. Materials and Methods

### 2.1. Participants and Procedures

In this observational study, patients with PD were consecutively recruited at the outpatient clinic of the Parkinson’s Disease Center of University Hospital of Rome Tor Vergata. All patients received a diagnosis of idiopathic PD according to the Movement Disorder Society’s criteria [23] and were required to meet the following inclusion criteria: ongoing pharmacological anti-parkinsonian treatment; no cognitive impairment (with Mini-Mental State Examination [24] score ≥ 24); being able to report about their condition and symptoms; being able to give their consent to participate in the study. Exclusion criteria were: concomitant neurologic and/or psychiatric diseases; systemic or metabolic diseases; arthritis and/or arthrosis with severe motor impairment; recent orthopedic surgery or neurosurgery and any condition interfering with the Global Mobility Task (GMT) [25] performed.

After neurological examination, all participants answered the NoMoFa questionnaire and subsequently underwent the GMT in ON state.

The study procedures were approved by the local Ethics Committee and were conducted in accordance with the Declaration of Helsinki. All participants provided signed informed consent.

### 2.2. Clinical Evaluation

Disease duration was obtained from patients’ clinical records, and levodopa-equivalent daily dose (LEDD) at the time of evaluation was computed for each participant. Disease severity was rated with the modified Hoehn and Yahr’s (H&Y) scale [26] (range 1–5), and motor examination was performed with the Unified Parkinson’s Disease Rating Scale part III (UPDRS-III) and part IV (UPDRS-IV) [27].

### 2.3. Measures

#### 2.3.1. Non-Motor Fluctuation Assessment Questionnaire

Non-motor symptoms and non-motor fluctuations were assessed with the NoMoFa questionnaire. This self-report instrument comprises 27 items, and each refers to common NMS of PD (e.g., difficulty planning an activity, hopelessness/sadness, sleepiness, excessive sweating, etc.; a complete list of the assessed symptoms is reported in Table 1). For each symptom, the respondent indicates: (1) whether he experienced or not that symptom in the last two weeks, (2) the severity of the symptom (on a three point scale, with 1 = mild, 2 = moderate, 3 = severe), and (3) whether the symptom was worse when levodopa was working (ON), when levodopa was not working (OFF), or it simply occurred with no difference in ON and OFF conditions. Three subscores can be computed, namely the total NMF subscore ON given by the sum of the severity scores for the symptoms only occurring in the ON state, the total NMF subscore OFF obtained by summing the severity scores for the symptoms in OFF state, and the total static NMS, which is the sum of severity scores for the symptoms occurring with no difference in ON and OFF states. The sum of the three subscores yields the total NoMoFa score, which is a synthetic measure of NMS. On the other hand, the ON and OFF subscores account for the presence and the intensity of NMF. In this study, NMF subscore ON ≥ 1 or NMF subscore OFF ≥ 1 were considered as indicative of the presence of NMF, and the number of ON, OFF, and static NMS were obtained by counting the items endorsed for each condition (ON state, OFF state, and no difference).

In the original validation study [22], the NoMoFa showed good reliability (Cronbach’s *α* = 0.89) and convergent validity with other measures of disease severity pertaining to both motor and non-motor symptoms (including UPDRS scores and quality of life as measured with the 8-item Parkinson’s Disease Questionnaire [28]).

#### 2.3.2. Global Mobility Task

Motor impairment was evaluated with the GMT in ON state. The task involves motor coordination, limb strength, and balance and consists of the five motor steps necessary to move from a supine to a prone position, described in detail by Peppe et al. [25]. For each step, the administering clinician rated the patient’s performance on a scale ranging from 0 to 4 (with 0 = the patient was completely autonomous, and 4 = the patient was completely dependent) and measured the time passing between the start and the end of the step. The maximum time allowed for a single step is 60 s, and the maximum total score for the task is 20. Higher scores and times correspond to worse motor impairment.

The GMT demonstrated excellent internal consistency in PD patients and healthy elderly (with *α* = 0.94 for the total score, and *α* = 0.94 for total time) and was shown to be significantly associated with PD severity as rated with the H&Y scale [25].

### 2.4. Statistical Analyses

All descriptive and inferential statistical analyses were conducted with IBM SPSS software (version 25.0, IBM Corp., Armonk, NY, USA). Categorical variables were summarized with counts and percentages with 95% confidence intervals, and means and standard deviations or medians and interquartile ranges were obtained for numerical variables according to data distribution.

The significance of the associations between categorical variables was examined with chi-square (*χ*^2^) tests, whereas group differences (patients with non-motor fluctuations vs. patients without non-motor fluctuations and females vs. males) for numerical variables were evaluated with Mann–Whitney *U*-test. Spearman’s rho (*ρ*) rank correlation coefficients were computed to explore the bivariate relationships between NoMoFa scores with patients’ demographic and clinical data and motor impairment.

For all the analyses conducted, *p*-values below 0.05 were considered statistically significant.

## 3. Results

### 3.1. Sample’s Characteristics and Frequency of NMF

Thirty patients were enrolled for the study, and five patients were excluded because of orthopedic conditions (*n* = 3) or recent neurosurgery (*n* = 2). Hence, the final sample included 25 patients. Participants’ demographic and clinical characteristics and mean NoMoFa and GMT scores are reported in Table 1. According to H&Y scores, disease severity was mostly mild to moderate, with no patients showing severe PD symptoms (H&Y scores above 3). Sixteen patients (64%, 95 CI: 42.6–81.3) experienced motor fluctuations (i.e., UPDRS-IV score ≥ 1), and mean UPDRS-III and UPDRS-IV scores were 21 ± 9.1 and 4.3 ± 4.1, respectively.

Overall, all the patients enrolled in the study reported NMS, and eight of them (32%, 95 CI: 15.7–53.6) also reported NMF. The mean number of NMS in ON and OFF states were, respectively, 1.5 ± 0.6 (median, 0; IQR, 2) and 0.5 ± 0.2 (median, 0; IQR, 1), whereas the mean number of static NMS was 7.6 ± 0.9 (median, 7; IQR, 7). In total, the mean number of reported NMS was 9.6 ± 1.0 (median, 9; IQR, 8). The frequencies of static NMS and NMS occurring in ON and OFF states are presented in Table 2. Notably, more than half of the patients reported word-finding difficulties, restlessness, sadness, poor short-term memory, fatigue, excessive daytime sleepiness, pain, urinary urgency, and constipation, and among them, a significant proportion of those experiencing word-finding difficulties, sadness, fatigue, sleepiness, pain, and urinary urgency reported that these symptoms were fluctuating.

### 3.2. Associations between NMF, Clinical Data, Motor Impairment, and Motor Fluctuations

No significant associations were found between the presence of NMF and sex (*χ*^2^ = 0.03, *p* = 0.86), disease severity as rated by the H&Y scale (*χ*^2^ = 1.74, *p* = 0.42), or motor fluctuations (*χ*^2^ = 2.82, *p* = 0.09), and no differences in age were observed between patients with and without NMF (*U* = 58.5, *p* = 0.58). Conversely, patients with NMF predictably reported a higher number of NMS in ON state (*U* = 17.0, *p* < 0.01) and in OFF state (*U* = 17.0, *p* < 0.01) as well as a higher total number of NMS (*U* = 20.5, *p* < 0.01), including both fluctuating and non-fluctuating NMS.

Correlation coefficients between NoMoFa scores and PD duration, LEDD, disease severity (H&Y score), motor disability (UPDRS-III score), and motor impairment (GMT score and time) are reported in Table 3. Overall, no relationships were observed between the intensity of NMF and patients’ clinical characteristics or motor impairment assessed with the GMT. Conversely, the NMS static subscore and the NoMoFa total score were both positively associated with the GMT scores, and the NoMoFa total score and the GMT performance were also significantly correlated with both H&Y and UPDRS-III scores but not with the UPDRS-IV score.

## 4. Discussion

This study preliminarily investigated the impact of fluctuating and non-fluctuating NMS, evaluated by using the NoMoFa questionnaire, on motor functioning and fluctuations as assessed by the GMT and UPDRS III and IV in a sample of PD patients in pharmacological treatment. Consistently with the previous literature, the results confirmed the high frequency of NMS reported by PD patients in the H&Y stage between 1 and 3 (mild to moderate severity) and also showed the presence of NMF in one-third of the investigated sample. The novel and main finding of the present study is the significant correlation between NMS, NMF, and motor impairment and performance as measured by the GMT and UPDRS-III. Notably, NMF did not correlate with motor fluctuations as assessed by the UPDRS-IV. In addition, all the PD patients involved in this study reported one or more than one NMS, and the one-third of them reporting NMF also exhibited a higher number of NMS. Somewhat consistently with previous research [6,14], cognitive and neuropsychiatric complaints (including difficulties in word finding and short-term memory, sadness and hopelessness, and restlessness as well as fatigue and excessive sleepiness) were reported most frequently along with pain and autonomic dysfunctions such as constipation and urinary urgency both as static NMS or as NMF. In contrast with previous evidence [14,29], NMF mostly occurred in the ON state in our sample, and we did not observe an association between sex and the presence of NMF. Moreover, and in line with the results already reported by Storch et al. [17], no significant relationships were found between NMF and disease severity, motor impairment, or motor fluctuations, whereas total NMS showed significant correlations with the UPDRS-III and the H&Y scores, and both static NMS and total NMS were strongly correlated with the GMT performance (which in turn was shown to be strongly associated with clinician-rated motor symptoms and overall disease severity). These findings further highlight the impact of non-motor manifestations on motor functioning and their relevance in the disease’s overall clinical management. Consistently, our results also showed that a simple structured motor task such as the GMT may help identifying the effect of NMS and NMF on motor performance in patients with PD, thus supporting the importance of including NMS assessment in the clinical monitoring of PD patients. Somewhat unexpectedly, but in line with recent reports [30,31], the use of anti-parkinsonian treatments and LEDD had no association with NMS. Accordingly, as further supported by their lack of association with motor fluctuations observed in this study, NMS and their fluctuations might be not intrinsically related to the dopaminergic action of pharmacotherapy, representing instead an inherent characteristic of PD whose mechanisms may be, at least in part, different from those responsible for motor fluctuations. Hence, non-pharmacological interventions such as neurorehabilitation and behavioral adaptation strategies [32] might play a fundamental role in the management of non-motor manifestations and should be tailored to specific patients’ needs and symptoms. Indeed, adapted cognitive-behavioral therapy was shown to be effective for psychiatric NMS of PD [33,34], and preliminary findings highlight its potential to also positively impact motor symptoms [35]. To optimize the delivery of these treatments, precisely identify their clinical targets, and further enhance their effectiveness, a thorough evaluation of NMS considering their relative severity and their pattern of occurrence is critical and might be routinely conducted. However, although several studies already addressed relevant issues concerning NMS and NMF, such as their prevalence and possible treatment strategies (including the adjustment of dopaminergic therapy as well as non-pharmacological strategies such as deep-brain stimulation) [36,37], they mostly relied on non-specific assessment procedures (e.g., open questions during clinical interview [16], semi-structured interviews [38], symptoms diaries [39], and repeated administrations of the Non-Motor Symptoms Scale [40]). In fact, self-report questionnaires showed higher sensitivity in the identification of both motor and non-motor wearing-off in comparison with routine clinical interviewing [41], and notably, the NoMoFa is the first dedicated instruments for the assessment of NMF. Given its feasible format and its sound psychometric properties, it might represent a suitable solution for the systematic study of non-motor manifestations in both research and clinical settings. To our knowledge, this is the first clinical research employing the NoMoFa for the assessment of NMS and NMF besides the first validation study [22], and our results further confirm its promising potential. In particular, its use along with clinical examination and task-based motor assessment allows drawing a comprehensive synthesis of the patient’s condition that might guide the choice of personalized treatments and the delivery of tailor-made rehabilitation plans. Considering the recent emphasis on personalized care in PD [42,43], such extensive assessment strategies might be highly recommended both in the initial contact and in the subsequent follow-up monitoring. Indeed, given the well-documented burden of NMS and their fluctuations on patients’ quality of life, activity of daily living, and adherence to treatments, they may also be considered along with motor symptoms as primary endpoints in clinical trials and real-world studies assessing the effectiveness of traditional and new therapeutic strategies, including pharmacological and non-pharmacological treatments.

As a whole, this pilot study showed that NMS are likely to be strongly associated with motor performance in PD but also supports the hypothesis that the mechanisms responsible for NMF and motor fluctuations might be heterogeneous, and thus, these two manifestations might require different treatment approaches due to different neurotransmitter systems impairment. Neural systems mainly involved in motor and non-motor manifestations are schematically illustrated in the Graphical Abstract, which also highlights the importance of monitoring fluctuations since treatment plans can be personalized according to patients’ symptoms and need of care. Future research should seek to further explore this compelling hypothesis given that the evidence concerning specific treatments for NMS is still limited [44,45,46]. Specifically, given the purported early appearance of non-motor manifestations and the potential of incurring in NMS in the early stages of the disease, studies evaluating NMS-related biomarkers are highly warranted [47] to shed light on their role in the disease pathophysiology and possibly identify effective clinical strategies to counteract the burden of NMS.

However, several limitations of our study should be critically considered in evaluating the relevance of the reported findings. First, considering the preliminary nature of the study, we relied on a limited sample size of mild-to-moderate PD patients. In particular, although only participants with no cognitive decline according to their clinical records were included, structured cognitive assessment was not carried out, and hence, it was not possible to examine the relationship between NMS and NMF and overall cognitive performance already observed in previous studies [48] or the specific impact of cognitive functioning on NMS severity. Furthermore, the study design was cross-sectional, and causal inferences cannot be made about the direction of the associations found between the investigated variables. Finally, the assessment of NMS and NMF solely relied on patients’ responses to the NoMoFa obtained during clinical examination. Despite the evidence on the NoMoFa’s validity and reliability, retrospective judgment might yield a biased evaluation of the symptoms’ occurrence and should ideally be complemented with continuous monitoring in daily life using ecological momentary assessment [49]. Subsequent studies might adopt such multi-method assessment strategies along with easy-to-administer, well-established cognitive measures such as the Montreal Cognitive Assessment (MoCA) [50] to overcome some of the above-reported shortcomings of our research.

## 5. Conclusions

In conclusion, this observational study showed that up to 32% of a sample of patients with PD under pharmacological treatment reported fluctuations in their NMS. NMF measured by the NoMoFa showed no significant associations with disease severity, motor impairment, or motor fluctuations, whereas static and total NMS were shown to be strongly correlated with motor performance as assessed by the GMT. In addition, no relationships were observed between LEDD and static or fluctuating NMS. Altogether, these findings confirm that NMS and their fluctuations are relevant aspects to take into account in PD management and suggest that their treatment might require specific, tailored pharmacological and non-pharmacological strategies. In this context, the NoMoFa and the GMT represent useful instruments that may be integrated in routine assessment and monitoring of PD to enhance personalized care. Future research should employ dedicated assessment procedures such as the NoMoFa to collect systematic evidence concerning NMS and NMF, and further studies conducted on larger samples with robust longitudinal designs are needed to confirm the consistency of our findings and their relevance for clinical practice.

## Figures and Tables

**Table 1 jpm-13-00242-t001:** Participants’ (N = 25) characteristics and mean NoMoFa and GMT scores.

Variable	N (%)	Mean (SD)
Sex		
Female	10 (40.0)	
Male	15 (60.0)	
Age		66.9 (10.3)
Disease duration (years)		6.3 (4.1)
Hoehn and Yahr score		2.2 (0.7)
1–1.5	6 (24.0)	
2–2.5	11 (45.0)	
3	8 (32.0)	
LEDD (mg)		581.6 (234.1)
UPDRS-III		21.0 (9.1)
UPDRS-IV		4.3 (4.1)
Motor fluctuations		
Yes	16 (64.0)	
No	9 (36.0)	
Non motor fluctuations		
Yes	8 (32.0)	
No	17 (68.0)	
NoMoFa NMF ON		2.6 (5.0)
NoMoFa NMF OFF		1.0 (2.7)
NoMoFa NMS static		14.4 (9.6)
NoMoFa total		18.0 (10.7)
GMT score		6.8 (6.6)
GMT time (seconds)		77.6 (99.8)

Note. LEDD, levodopa-equivalent daily dose; UPDRS-III and UPDRS-IV, Unified Parkinson’s Disease Rating Scale part III and part IV; NoMoFa, Non-Motor Fluctuations Assessment questionnaire; NMF ON, ON-state non-motor fluctuations subscore; NMF OFF, OFF-state non-motor fluctuations subscore; GMT, Global Mobility Task.

**Table 2 jpm-13-00242-t002:** Frequencies of total, fluctuating (ON and OFF), and static non-motor symptoms.

NMS	Total Frequency NMS (N (%))	ON NMS	OFF NMS	Static NMS
Loss of train of thought	8 (32.0)	-	-	8 (32.0)
Distraction (difficulty completing task)	6 (24.0)	1 (4.0)	-	5 (20.0)
Difficulty planning an activity	9 (36.0)	2 (8.0)	2 (8.0)	5 (20.0)
Confusion	3 (12.0)	2 (8.0)	-	1 (4.0)
Word finding difficulty	14 (56.0)	3 (12.0)	-	11 (44.0)
Excessive worry	12 (48.0)	-	1 (4.0)	11 (44.0)
Fear (feeling scared)	6 (24.0)	-	2 (8.0)	4 (16.0)
Restlessness	14 (56.0)	1 (4.0)	-	13 (52.0)
Sadness/hopelessness	13 (52.0)	2 (8.0)	2 (8.0)	9 (36.0)
Loneliness/isolation	4 (16.0)	-	1 (4.0)	3 (12.0)
Hallucinations	3 (12.0)	-	-	3 (12.0)
Poor decision making	1 (4.0)	-	-	1 (4.0)
Impulsiveness	4 (16.0)	1 (4.0)	-	3 (12.0)
Compulsions/uncontrollable urges	5 (20.0)	1 (4.0)	-	4 (16.0)
Poor short-term memory	16 (64.0)	1 (4.0)	-	15 (60.0)
Difficulty handling stressful situations	3 (12.0)	1 (4.0)	-	2 (8.0)
Apathy/loss of interest	8 (32.0)	3 (12.0)	-	5 (20.0)
Low energy/fatigue	17 (68.0)	5 (20.0)	-	12 (48.0)
Excessive daytime sleepiness	16 (64.0)	4 (16.0)	1 (4.0)	11 (44.0)
Pain	19 (76.0)	2 (8.0)	2 (8.0)	15 (60.0)
Altered sensations	1 (4.0)	1 (4.0)	-	-
Shortness of breath	7 (28.0)	1 (4.0)	1 (4.0)	5 (20.0)
Changes in vision	7 (28.0)	1 (4.0)	-	6 (24.0)
Excess sweating	10 (40.0)	1 (4.0)	-	9 (36.0)
Palpitations	1 (4.0)	-	-	1 (4.0)
Urinary symptoms	18 (72.0)	3 (12.0)	-	15 (60.0)
Constipation	14 (56.0)	1 (4.0)	-	13 (52.0)

**Table 3 jpm-13-00242-t003:** Spearman’s correlation coefficients between NoMoFa subscores and total score, clinical data, and GMT scores.

Variable	1.	2.	3.	4.	5.	6.	7.	8.	9.	10.	11.
1. NoMoFa NMF ON	1										
2. NoMoFa NMF OFF	0.50 *	1									
3. NoMoFa NMS	−0.23	0.01	1								
4. NoMoFa Total	0.37	0.43 *	0.78 ***	1							
5. Disease duration	−0.15	0.07	0.00	−0.02	1						
6. LEDD	−0.10	−0.11	−0.17	−0.18	0.49 *	1					
7. H&Y	0.09	0.15	0.38	0.46 *	0.38	0.25	1				
8. UPDRS-III	0.20	0.15	0.33	0.45 *	0.22	0.17	0.89 ***	1			
9. UPDRS-IV	0.11	0.33	0.02	0.15	0.65 ***	0.40 *	0.55 **	0.50*	1		
10. GMT score	−0.09	0.20	0.65 ***	0.58 **	0.09	−0.05	0.66 ***	0.59 **	0.32	1	
10. GMT time	0.02	0.27	0.68 ***	0.66 ***	0.14	−0.07	0.71 ***	0.66 ***	0.37	0.95 ***	1

Note. LEDD, levodopa-equivalent daily dose; UPDRS-III, Unified Parkinson’s Disease Rating Scale part III; NoMoFa, Non-Motor Fluctuations Assessment questionnaire; NMF ON, ON-state non-motor fluctuations subscore; NMF OFF, OFF-state non-motor fluctuations subscore; GMT, Global Mobility Task. * *p* < 0.05; ** *p* < 0.01; *** *p* < 0.001.

## Data Availability

The data that support the results reported in this study are available from the corresponding author upon reasonable request.

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
