# Peer review of "Both Motor and Non-Motor Fluctuations Matter in the Clinical Management of Patients with Parkinson’s Disease: An Exploratory Study"

_jpm, 2023, doi:10.3390/jpm13020242_

Round 1
Reviewer 1 Report
The article "Both motor and non-motor fluctuations matter in the clinical
management of patients with Parkinson’s disease: an exploratory study" is well organized by authors and can be accepted in the JPM after minor revision.
I have two questions.
1. During such an observational study 'have you got any dementia symptoms in any Parkinson's patient? If yes, kindly reflect it in the paper. Since dementia is also one of the symptoms along with motor fluctuation in PD.
2. What type of treatments were most suitable for patients with NMS and NMF? That should be presented to enlighten the manuscript.
Author Response
We thank the reviewer for his/her keen and helpful observations. Our responses to the reviewer’s specific questions are reported below (reviewer's comments are reported in bold). Changes were made in the revised manuscript file using the Track Changes function of Microsoft Word and were highlighted in yellow. New references were highlighted in yellow in the references list.
1) During such an observational study 'have you got any dementia symptoms in any Parkinson's patient? If yes, kindly reflect it in the paper. Since dementia is also one of the symptoms along with motor fluctuation in PD.
As mentioned in the manuscript, patients with cognitive impairment or dementia (i.e., MMSE score below 24 reported in their recent clinical record) were initially excluded. Although structured cognitive assessment was not carried out during the study protocol, no patient involved in the study showed any sign of cognitive impairment. We acknowledge that this made it impossible to ascertain the relationships between dementia symptoms and both motor and non-motor fluctuations in our study and thus we add this point in the revised manuscript in the paragraph dealing with limitations in the Discussion section.
2) What type of treatments were most suitable for patients with NMS and NMF? That should be presented to enlighten the manuscript.
Pharmacological treatment was not modified during the study and unfortunately we did not investigate the longitudinal trend of NMS and NMF in our sample. Hence, we were not able to report the impact of different treatment strategies. As discussed in the paper, definitive guidelines for the management of NMS and NMF are not available, but systematic research might shed light on feasible solutions. We further discuss this point in the revised manuscript by highlighting the importance of studies investigating biomarkers for the detection and for set specific treatment for NMS and suggesting that NMS and NMF assessment might be considered as endpoints in future clinical trials on PD patients.
Reviewer 2 Report
The Manuscript by Carpi et al. (Manuscript ID: jpm-2149822) titled "Both motor and non-motor fluctuations matter in the clinical management of patients with Parkinson’s disease: an exploratory study" aimed to investigate the presence of Non-motor symptoms (NMS) and NMS fluctuations (NMF) in patients with Parkinson’s disease. The authors used the Non-Motor Fluctuation Assessment questionnaire (NoMoFa) to evaluate their associations with disease characteristics and motor impairment. The authors found that one-third of the patients displayed NMF, and patients with NMF displayed a higher number of NMS. The authors concluded that NMF is frequently reported by PD patients and linked with an augmented NMS. I think the work is interesting. I have some comments that may help to improve the quality of the manuscript as follows:
- Since Parkinson’s disease has been associated with cognitive impairment at the later stage of PD, authors should provide more information on the cognitive status of PD patients. The cognitive status of the patients may have an effect on NMS and/or NMF. Do the authors have any comments on this?
- Do authors have information about the MoCA scores of PD patients and its correlation with NMS?
- Authors are suggested to add a schematic representation to understand the overall conclusion of the manuscript.
- Authors should discuss the importance of their study in biomarker and/or therapeutic developments in Parkinson’s disease in the manuscript.
Author Response
We thank the reviewer for his/her useful comments and observations. Our point-by-point responses to the reviewer’s specific questions are reported below (reviewer's comments are reported in bold). Changes were made in the revised manuscript file using the Track Changes function of Microsoft Word and were highlighted in yellow. New references were highlighted in yellow in the references list.
Since Parkinson’s disease has been associated with cognitive impairment at the later stage of PD, authors should provide more information on the cognitive status of PD patients. The cognitive status of the patients may have an effect on NMS and/or NMF. Do the authors have any comments on this?
We thank the reviewer for pointing out this relevant issue. We do agree that the relation between non-motor manifestations and cognitive impairment should be thoroughly investigated. Unfortunately, complete cognitive assessment was not conducted in our study and thus it was not possible to examine the associations between cognitive performance and NMS. In addition, our sample might not have been adequate to detect such relationship, since all the included participants did not show cognitive impairments according to clinical records (MMSE score above 24) nor severe PD symptoms. In the revised manuscript, we highlighted this limitation in the Discussion section.
Do authors have information about the MoCA scores of PD patients and its correlation with NMS?
As mentioned above, the only available cognitive measure was the MMSE score reported in the patients’ clinical record. We agree with the reviewer that the MoCA might have provided a practical solution to rapidly collect data on patients’ performance in the major cognitive domains and remarked that in the revised manuscript.
Authors are suggested to add a schematic representation to understand the overall conclusion of the manuscript.
A graphical abstract illustrating the results of the study was submitted along with the paper. As suggested by the reviewer, in the revised version of the manuscript we added the illustration (as Figure 1) to schematically summarized our conclusions.
Authors should discuss the importance of their study in biomarker and/or therapeutic developments in Parkinson’s disease in the manuscript.
We thank the reviewer for this relevant suggestion. We indeed reflected upon this issue in conceptualizing our work. In the revised manuscript, we made it explicit by highlighting the relevance of studies investigating biomarker of NMS in the early identification of PD. In addition, besides reporting the possible role of neurorehabilitation and behavioral intervention as treatment strategies, we further stressed that NMS and NMF assessment might be considered as endpoints in future clinical trials in PD patients in order to identify most feasible treatment strategies to reduce their burden on patients’ well-being and quality of life.